# In-Context Compositional Learning via Sparse Coding Transformer

**Wei Chen, Jingxi Yu, Zichen Miao, Qiang Qiu**
Purdue University, IN, USA
{chen2732, yu667, miaoz, qqiu}@purdue.edu

## Abstract

Transformer architectures have achieved remarkable success across language, vision, and multimodal tasks, and there is growing demand for them to address in-context compositional learning tasks. In these tasks, models solve the target problems by inferring compositional rules from context examples, which are composed of basic components structured by underlying rules. However, some of these tasks remain challenging for Transformers, which are not inherently designed to handle compositional tasks and offer limited structural inductive bias. In this work, inspired by the principle of sparse coding, we propose a reformulation of the attention to enhance its capability for compositional tasks. In sparse coding, data are represented as sparse combinations of dictionary atoms with coefficients that capture their compositional rules. Specifically, we reinterpret the attention block as a mapping of inputs into outputs through projections onto two sets of learned dictionary atoms: an *encoding dictionary* and a *decoding dictionary*. The encoding dictionary decomposes the input into a set of coefficients, which represent the compositional structure of the input. To enhance structured representations, we impose sparsity on these coefficients. The sparse coefficients are then used to linearly combine the decoding dictionary atoms to generate the output. Furthermore, to assist compositional generalization tasks, we propose estimating the coefficients of the target problem as a linear combination of the coefficients obtained from the context examples. We demonstrate the effectiveness of our approach on the S-RAVEN and RAVEN datasets. For certain compositional generalization tasks, our method maintains performance even when standard Transformers fail, owing to its ability to learn and apply compositional rules.

## 1 Introduction

Recent advancements in artificial intelligence (AI) have led to significant breakthroughs in various domains [6, 11, 17, 26]. Models such as large-scale Transformers have demonstrated remarkable capabilities in natural language understanding, image classification, and multimodal reasoning. However, despite these successes, solving in-context compositional learning tasks remains a major challenge [20]. As illustrated in Figure 1, such tasks involve data composed of basic components arranged by underlying compositional rules, requiring models to infer and transfer these structural patterns from context examples while achieving good representation and generalization.

Transformers primarily rely on dense attention mechanisms [26] without an explicit framework for representing compositional rules. As a result, they struggle to capture structured relationships and lack an effective mechanism to transfer these inferred rules across examples. The absence of structural inductive bias limits their ability to generalize in tasks that demand compositional understanding.

In this paper, we extend the attention mechanism by explicitly encoding compositional rules, drawing inspiration from the principles of sparse coding. In sparse coding [14], signals are expressed as sparse combinations of basic elements, with the resulting coefficients capturing the compositional structure

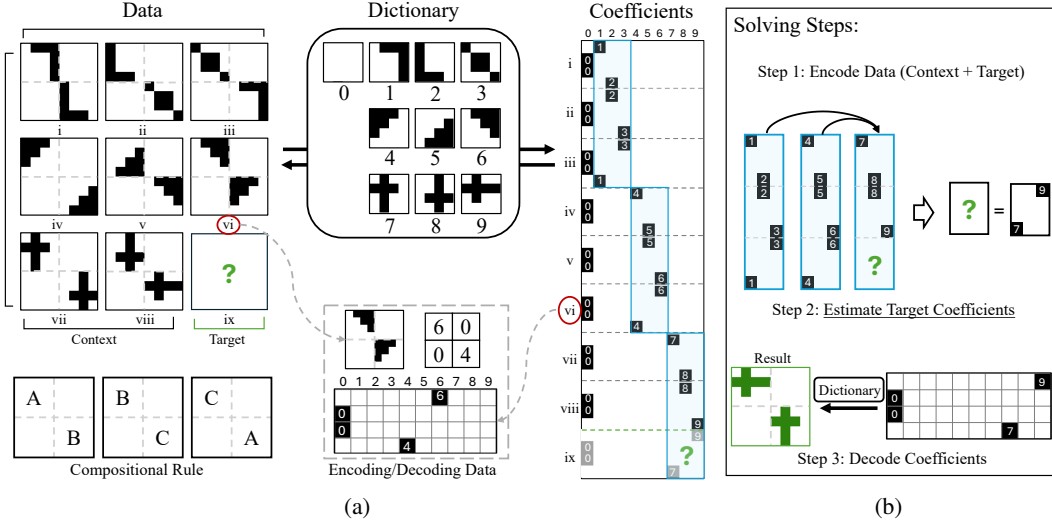

Figure 1: Illustration of the in-context compositional learning task. The input data includes both the context tasks and the target task. The goal is to solve the target task by inferring and applying the compositional rule observed in the context tasks. **(a)** Applying the principles of sparse coding to represent the data. Given a dictionary, the input data can be *sparsely* represented using a set of coefficients that encode underlying *compositional rules*. **Encoding/decoding data:** An example of one task is composed of four elements from the dictionary, with indices "6, 0, 0, 4." After one-hot embedding, we obtain a $4 \times 10$ matrix, where each nonzero entry corresponds to a specific element in the dictionary. By stacking all 9 examples, we obtain a $36 \times 10$ matrix representing the coefficients. **Compositional rules:** Each row of the input data follows an underlying pattern. If the first two shapes are constructed as $(A, \emptyset, \emptyset, B)$ and $(B, \emptyset, \emptyset, C)$, where $A$, $B$, and $C$ correspond to unique elements in the dictionary, $\emptyset$ means an empty shape, then the third shape should be $(C, \emptyset, \emptyset, A)$. **(b)** Representing the compositional rules as coefficients provides an effective way to estimate the coefficients of the target task from those of the context tasks. Once inferred, these coefficients can be decoded into the final output using the dictionary. Details of this task are described in Section 3.

of the signal. Specifically, as shown in Figure 2, we reinterpret the attention mechanism as a mapping of inputs into outputs through projections onto two sets of learned dictionary atoms: an *encoding dictionary* and a *decoding dictionary*. The encoding dictionary decomposes the input into a set of coefficients, which represent the compositional structure of the input. The coefficients are then used to linearly combine the decoding dictionary atoms to generate the output.

In the attention mechanism [26], the attention map is generated by computing the inner product between inputs transformed by the query and key matrices. In contrast, our approach reinterprets this process as projecting the input onto a learned dictionary, *i.e.*, encoding dictionary, parameterized by the query and key matrices to obtain the coefficients. To enhance structured representations, we introduce sparsity into the coefficients, allowing them to explicitly represent the compositional rules inherent in the input. These sparse coefficients are then used to combine another dictionary, *i.e.*, decoding dictionary, parameterized by the value matrix, to generate the final output.

By projecting the input of both the context and target tasks onto a shared encoding dictionary to obtain their respective coefficients, we can effectively infer the compositional rules of the target tasks. Inspired by the lifting scheme [22], we estimate coefficients of the target task through a simple linear combination of the context task coefficients.

We first assess the effectiveness of our method on a toy example with a simple compositional rule, demonstrating that our approach successfully learns and generalizes the rule, whereas the standard Transformer fails in this case. The results are shown in Figure 3. We then evaluate our method on the in-context compositional learning dataset, such as S-RAVEN [20] and RAVEN [29]. Our approach consistently outperforms standard Transformer baselines. These results indicate that integrating the attention mechanism with sparse coding enhances the ability of models to learn and apply compositional rules.

We summarize our contributions as follows:

- We reformulate the attention mechanism, inspired by sparse coding, as a mapping of inputs to outputs via projections onto two learned dictionaries: an encoding dictionary and a decoding dictionary.
- We explicitly represent inputs as sparse combinations of the encoding dictionary to encode compositional rules.
- We enable effective transfer of compositional rules across tasks by estimating target coefficients via a simple linear combination of context coefficients.
- We demonstrate the effectiveness of our approach on in-context compositional learning tasks, maintaining good performance even in cases where standard Transformers fail.

## 2  Method

In this section, we first outline the problem setting of in-context compositional learning and then introduce our framework, inspired by sparse coding, which reformulates the Transformer architecture to better capture compositional structure.

### 2.1  Preliminary

**Problem formulation.**  We define the in-context compositional learning task as learning a function purely from demonstrations provided within a context window. Inspired by the RAVEN dataset [29], we consider a setting where the model is given $L-1$ structured example (the *context*) and predicts the $L^{\text{th}}$ one (the *target*). We illustrate this task in Figure 1.

Assume each example $x_i \in \mathcal{X}$ is governed by a latent compositional rule $\mathcal{R}$. Let the **context set** be:

$$\mathcal{C} = \{x_1, x_2, \ldots, x_{L-1}\} \subset \mathcal{X}^{L-1}. \tag{1}$$

The model must produce $\hat{x}_L \in \mathcal{X}$ such that $\hat{x}_L = f(\mathcal{C})$, where $f$ is a learned model conditioned on the context $\mathcal{C}$. The goal is to minimize the expected error over a distribution of tasks:

$$\min_f \ \mathbb{E}_{\mathcal{C},x_L} \left[ \ell \left( f(\mathcal{C}), x_L \right) \right], \tag{2}$$

where $\ell$ is a task-specific loss function. To emphasize in-context compositional learning, the tasks in the distribution $\mathcal{D}$ are constructed such that: (1) Each rule $\mathcal{R}$ is composed from a finite set of primitive operations $\mathcal{P}$. (2) Test-time tasks involve novel combinations of primitives not seen during training, *i.e.*, $\mathcal{R}_{\text{test}} \notin \text{span}(\mathcal{R}_{\text{train}})$. This setting evaluates the model's ability to infer latent rules purely from examples and apply them to unseen inputs in a compositional manner, mimicking human inductive reasoning in Raven's Progressive Matrices.

**Sparse coding.**  Sparse coding represents signals using linear combinations of an overcomplete dictionary $\mathbf{D} \in \mathbb{R}^{m \times d}$ (where $m > d$ is the number of atoms), and representing the signal as:

$$\mathbf{X} \approx \mathbf{SD}, \tag{3}$$

where $\mathbf{X} \in \mathbb{R}^{N \times d}$ is the input signal, $\mathbf{S} \in \mathbb{R}^{N \times m}$ is the sparse coefficient vector, which has only a few nonzero elements. To achieve sparsity, the common usage is the soft-thresholding function:

$$\text{prox}(\mathbf{S}) = \text{sign}(\mathbf{S}) \odot \max(|\mathbf{S}| - \xi, 0), \tag{4}$$

where $\odot$ is Hadamard product. It encourages sparsity by shrinking small values of $\mathbf{S}$ toward zero. Sparse coding is widely used in signal processing, machine learning, and neuroscience, providing efficient and interpretable representations of data.

### 2.2  Revisiting Transformer Blocks

**Multi-head attention (MHA).**  The attention layer [26] transforms the input sequence $\mathbf{X} \in \mathbb{R}^{N \times d}$ to the output sequence $\mathbf{O} \in \mathbb{R}^{N \times d}$, where $N$ denotes the sequence length, $d$ is the dimension of

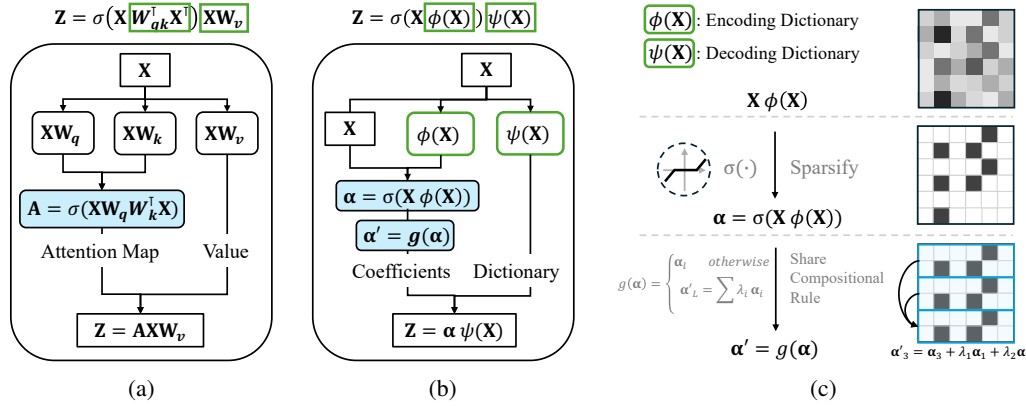

Figure 2: (a) The attention block produces the output as a linear combination of the value matrix, weighted by the attention map. (b) Our framework reformulates the attention mechanism: Outputs are constructed as sparse combinations of learned dictionary atoms, *i.e.*, *decoding dictionary* $\psi(\mathbf{X})$, and their coefficients $\boldsymbol{\alpha}$ represent compositional rules. (c) Details of our method: The coefficients $\boldsymbol{\alpha}$ are obtained by decomposing the input features over the *encoding dictionary* $\phi(\mathbf{X})$, and then achieving sparse representations with a nonlinear function $\sigma(\cdot)$. Since the coefficients of the target task only provide partial information about its compositional rule due to limited observations, we propose to estimate the coefficients of the target task $\boldsymbol{\alpha}_L$ as a simple linear combination of the context task coefficients, *i.e.*, $\boldsymbol{\alpha}' = g(\boldsymbol{\alpha})$. Further details are provided in Section 2.3.

input and output features. The attention layer projects the input using the corresponding projection matrices $\mathbf{W}_q, \mathbf{W}_k, \mathbf{W}_v \in \mathbb{R}^{d \times d}$, and calculates the attention map,

$$\mathbf{A} = \mathtt{ATTN}(\mathbf{X}) = \sigma(\mathbf{X}\mathbf{W}_q\mathbf{W}_k^\intercal\mathbf{X}^\intercal), \tag{5}$$

where $\sigma(\cdot) = \mathtt{softmax}(\cdot)$. The attention map $\mathbf{A} \in \mathbb{R}^{N \times N}$ captures the token-wise relationship by doing inner-product in a space transformed by $\mathbf{W}_q, \mathbf{W}_k$.

Multi-head attention extends this by allowing multiple attention mechanisms to work in parallel, with each head independently learning attention patterns. For $H$ attention heads, each attention head calculates attention maps as $\mathbf{A}^{(h)} = \sigma(\mathbf{X}\,\mathbf{W}_{qk}^{(h)}\mathbf{X}^\intercal)$, where $\mathbf{W}_{qk}^{(h)} = \mathbf{W}_q^{(h)}\mathbf{W}_k^{(h)\intercal}$ is corresponding to projection matrices $\mathbf{W}_q^{(h)}, \mathbf{W}_k^{(h)} \in \mathbb{R}^{d \times \frac{d}{H}}, h = 1, \cdots, H$. The multi-head attention represents as,

$$\mathtt{MHA}(\mathbf{X}) = \sum_{h=1}^{H} \mathbf{A}^{(h)}\mathbf{X}\mathbf{W}_{vo}^{(h)} = \sum_{h=1}^{H} \sigma(\mathbf{X}\,\mathbf{W}_{qk}^{(h)}\mathbf{X}^\intercal)\,\mathbf{X}\mathbf{W}_{vo}^{(h)}, \tag{6}$$

where $\mathbf{W}_{vo}^{(h)} = \mathbf{W}_v^{(h)}\mathbf{W}_o^{(h)\intercal}, \mathbf{W}_v^{(h)}, \mathbf{W}_o^{(h)} \in \mathbb{R}^{d \times \frac{d}{H}}, \mathbf{W}_{vo}^{(h)} \in \mathbb{R}^{d \times d}$.

While the MHA offers a form of learned localization via query-key similarity, it suffers from two fundamental limitations in compositional tasks:

- The use of the $\mathtt{softmax}$ function produces dense attention weights, resulting in indiscriminate global mixing of information. This lack of sparsity hinders the model from representing the compositional structure inherent in contextual tasks.

- There is no explicit mechanism for reusing local compositional rules. It struggles to disentangle meaningful subcomponents, limiting its capacity to generalize via transferring compositional rules.

## 2.3 Reformulate Transformer Using Sparse Coding

We propose to explicitly reinterpret the attention in the Transformer as a form of learned, sparse coding problem. We factorize MHA (6) as follows:

$$
\begin{aligned}
\texttt{MHA}(\mathbf{X}) &= \sum_{h=1}^{H} \sigma(\mathbf{X} \underbrace{\mathbf{W}_{qk}^{(h)}\mathbf{X}^{\intercal}}) \ \underbrace{\mathbf{X}\mathbf{W}_{vo}^{(h)}} \\
&= \sum_{h=1}^{H} \sigma(\mathbf{X} \ \underbrace{\phi^{(h)}(\mathbf{X})}_{\text{Encoding dictionary}} \ ) \ \underbrace{\psi^{(h)}(\mathbf{X})}_{\text{Decoding dictionary}} \ ,
\end{aligned}
\tag{7}
$$

where $\phi^{(h)}(\mathbf{X})$ and $\psi^{(h)}(\mathbf{X})$ generate a set of dictionary atoms conditioned on the input $\mathbf{X}$, $\phi^{(h)}(\cdot)$ and $\psi^{(h)}(\cdot)$ are the basis functions parameterized by $\mathbf{W}_{qk}^{(h)}$ and $\mathbf{W}_{vo}^{(h)}$. Our method is illustrated in Figure 2.

**Learned dictionary atoms.** Our method reformulates the attention mechanism as a composition over learned dictionary atoms to enable structured representations. Specifically, we introduce two sets of input-dependent dictionaries: $\phi(\mathbf{X})$ and $\psi(\mathbf{X})$, both parameterized by learnable functions of the input $\mathbf{X}$.

- The encoding dictionary $\phi(\mathbf{X})$ is used to extract coefficients by computing the product $\mathbf{X}\phi(\mathbf{X})$, which represents how the input $\mathbf{X}$ decomposed with respect to the learned dictionary atoms. These coefficients encode the combination rule underlying the input structure.
- The decoding dictionary $\psi(\mathbf{X})$ serves as a reconstruction dictionary that synthesizes the final output from the coefficients.

Both $\phi(\mathbf{X})$ and $\psi(\mathbf{X})$ are dynamic and data-dependent, allowing the model to adaptively learn dictionary atoms that best represent the compositional patterns in each input instance.

**Sparse coefficients.** The coefficients $\mathbf{X}\,\phi(\mathbf{X})$ encode the combination rule underlying the input structure. To enhance the model's capability to capture compositional structure, we apply sparsity-promoting nonlinearities $\sigma(\cdot)$, such as *soft-thresholding*, defined as $\text{prox}(x) = \text{sign}(x) \odot \max(|x| - \xi, 0)$ to introduce sparsity in coefficients $\boldsymbol{\alpha}$, where $\xi$ is the threshold for setting values to zero, *i.e.*,

$$
\boldsymbol{\alpha} = \sigma\left(\mathbf{X}\,\phi(\mathbf{X})\right).
\tag{8}
$$

Different from the attention map $\mathbf{A}$, which applies `softmax` operation, sparse coefficients preserve the most informative components while suppressing redundant interactions. The resulting representation is more structured and better aligned with the underlying compositional rules.

**Update coefficients of the target task.** By encoding the underlying compositional rule as sparse coefficients $\boldsymbol{\alpha}$, we aim to transfer this rule from context tasks to the target task. The coefficients of the target task encode only partial information about its compositional rule due to limited observations of itself. We can transfer the compositional rule to the target task by coefficient transfer.

To address this, we propose estimating the target coefficients based on those of the context tasks. Inspired by the lifting scheme [22], we devise a procedure that predicts the target task coefficients through a linear combination of the context task coefficients. Specifically, sparse coefficients $\boldsymbol{\alpha} \in \mathbb{R}^{N \times N}$ consist of contributions from both context and target tasks, with $L - 1$ portions derived from the context tasks and a single portion $\boldsymbol{\alpha}_L \in \mathbb{R}^{\frac{N}{L} \times N}$ corresponding to the target task. We can update the coefficients of target tasks $\boldsymbol{\alpha}_L$ by,

$$
\boldsymbol{\alpha}_L \leftarrow \boldsymbol{\alpha}_L + \sum_{i=1}^{L-1} \lambda_i \boldsymbol{\alpha}_i,
\tag{9}
$$

where $\lambda_i$ is the learnable parameter for combining the coefficients of context tasks. The coefficients of context tasks remain unchanged. We represent this operation with a function $g(\cdot)$,

$$
g(\boldsymbol{\alpha}) = \begin{cases} \boldsymbol{\alpha}_i & \text{context tasks,} \\ \boldsymbol{\alpha}_L + \sum_{i=1}^{L-1} \lambda_i \boldsymbol{\alpha}_i & \text{target task.} \end{cases}
\tag{10}
$$

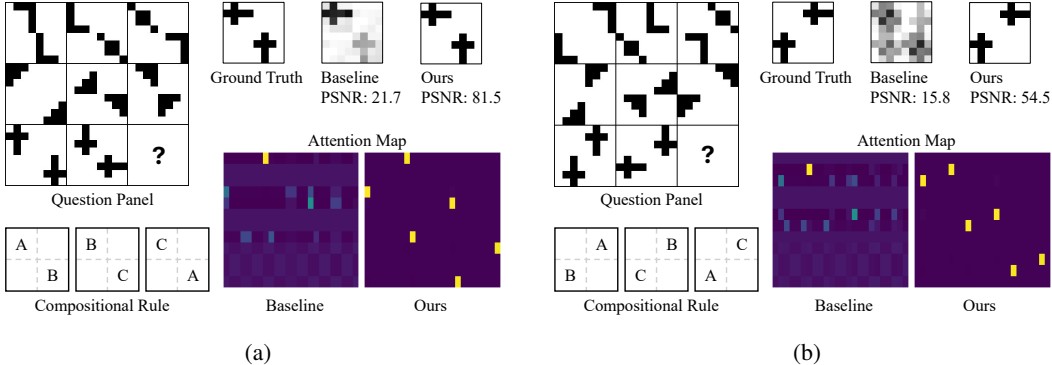

Figure 3: The effectiveness of sparse coefficients (attention map). Models are trained on setting (a) and tested on both setting (a) and novel setting (b), which has a different compositional rule. The baseline method, Transformer with standard MHA, produces blurry outputs due to dense coefficients, which lead to mixed and entangled results. In contrast, our sparse coefficients prevent this blurring and effectively transfer the construction rule from the context tasks to the target task. Further details are in Section 3.

This method is parameter-efficient. At each layer, there are $L - 1$ learnable parameters $\lambda_i$ corresponding to the number of context tasks, which remains relatively small compared to the overall parameter count of the Transformer blocks.

**Variation of basis functions.** This formulation allows us to explore various designs for the basis functions $\phi(\cdot)$ and $\psi(\cdot)$ to modulate the expressiveness of the model. A detailed discussion is provided in Appendix 8.

## 3 Discussion

We construct a synthetic dataset designed to evaluate in-context compositional learning. Each input consists of 9 panels. The panels are grouped such that panels 1–3, 4–6, and 7–9 share the same underlying compositional rule, as illustrated in Figure 3. The first two rows represent the context tasks, while the last row is the target task, which the model predicts based on the pattern observed in the first two examples.

**Compositional rules.** Each panel is an $8 \times 8$ binary image composed of four smaller basic shapes, arranged according to a predefined rule. For example, in Figure 3 (a), every three panels are composed of 3 basic shapes. Denoting these shapes as $A, B, C$, and use $\emptyset$ to represent an empty position, the panels are arranged from left to right and top to bottom as follows: $(A, \emptyset, \emptyset, B)$, $(B, \emptyset, \emptyset, C)$, and $(C, \emptyset, \emptyset, A)$. Similarly, the compositional rule of Figure 3 (b) is: $(\emptyset, A, B, \emptyset)$, $(\emptyset, B, C, \emptyset)$, and $(\emptyset, C, A, \emptyset)$. The basic shapes are chosen from a set of 16 elements, allowing for about $P(16, 9) = \frac{16!}{(16-9)!} \approx 4 \times 10^9$ distinct panel configurations. Details of the experimental setting are described in Appendix 7.

**Learning configures.** A single-layer Transformer block, containing only an attention layer, is trained to predict the target panel given the 8 context panels. The model is trained with a mean squared error (MSE) loss. The target panel is masked in the input, and the model is optimized to reconstruct it from the context examples. During training, the model is exposed to data generated under one compositional rule and evaluated on test data generated under a different rule to assess compositional generalization.

### 3.1 Effectiveness of Sparse Coefficients

We compare our approach with the baseline, where our method introduces sparsity in the coefficients. As shown in Figure 3, the baseline model with dense attention fails to predict the target panel on the test set and produces only blurry predictions on the training data. In contrast, through sparse attention and coefficient transfer, our method effectively infers and applies compositional rules to accurately predict the target panel on both training and test data, as illustrated in Figure 3 (a) and (b).

| Layers Training Tasks | 4 layers | | | 8 layers | | |
|---|---|---|---|---|---|---|
| | 10M | 20M | 40M | 10M | 20M | 40M |
| Transformer | $51.6\pm 1.3$ | $55.7\pm 1.5$ | $58.1\pm 1.4$ | $59.8\pm 1.4$ | $63.3\pm 1.9$ | $65.1\pm 4.3$ |
| HYLA [20] | $55\pm 2.1$ | $68.6\pm 1.5$ | $73.2\pm 0.6$ | $72.5\pm 6.6$ | $77.1\pm 3.4$ | $79.3\pm 1.8$ |
| Ours | $\mathbf{63.1}\pm 2.8$ | $\mathbf{73.9}\pm 3.8$ | $\mathbf{76.3}\pm 2.1$ | $\mathbf{72.6}\pm 3.9$ | $\mathbf{78.2}\pm 3.9$ | $\mathbf{82.7}\pm 2.5$ |

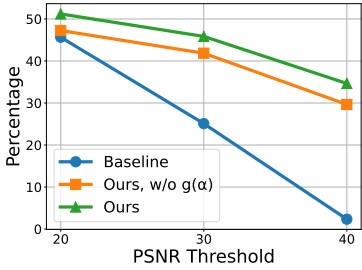

Figure 4: (Table) Accuracy comparison between our method and baseline methods on the Symbolic RAVEN (**S**-RAVEN) dataset. Our method consistently achieves higher accuracy than baselines. (Plot) Results on the **RAVEN** dataset. It shows the percentage of test samples with PSNR values exceeding a given threshold. At lower PSNR levels, the baseline method performs similarly to ours. However, for PSNR values above 40, the baseline achieves nearly **0** coverage, whereas our method retains over 30% of the samples.

## 3.2 Effectiveness of Coefficient Transfer

By representing an input $\mathbf{X}$ as $[\mathbf{X}_1, \cdots, \mathbf{X}_L]^{\mathsf{T}}$, where $\mathbf{X}_i, \forall i = 1, \cdots, L-1$ and $\mathbf{X}_L \in \mathbb{R}^{\frac{N}{L}\times d}$ are corresponding to context tasks and the target task, we have output according to (7),

$$\begin{bmatrix} \mathbf{Z}_1 \\ \vdots \\ \mathbf{Z}_L \end{bmatrix} = \begin{bmatrix} \sigma(\mathbf{X}_1\,\phi(\mathbf{X}))\,\psi(\mathbf{X}) \\ \vdots \\ \sigma(\mathbf{X}_L\,\phi(\mathbf{X}))\,\psi(\mathbf{X}) \end{bmatrix} = \begin{bmatrix} \boldsymbol{\alpha}_1\,\psi(\mathbf{X}) \\ \vdots \\ \boldsymbol{\alpha}_L\,\psi(\mathbf{X}) \end{bmatrix}. \tag{11}$$

We set $\mathbf{X}_L = \mathbf{0}$, where $\mathbf{0} \in \mathbb{R}^{\frac{N}{L}\times d}$ is a matrix with all zeros, since no observation for the target task.

**Baseline methods.** A standard Transformer with $\sigma(\cdot) = \texttt{softmax}(\cdot)$, produces coefficients $\boldsymbol{\alpha}_L = \sigma(\mathbf{X}_L\phi(\mathbf{X})) = \texttt{softmax}(\mathbf{0}) = \frac{1}{N}\mathbf{1}$, where $\mathbf{1} \in \mathbb{R}^{\frac{N}{L}\times d}$ is a matrix with all ones. It leads to the output of the target task as

$$\mathbf{Z}_L = \frac{1}{N}\mathbf{1}\psi(\mathbf{X}) = (\frac{1}{N}\mathbf{1}\mathbf{X})\mathbf{W}_v, \tag{12}$$

where $\frac{1}{N}\mathbf{1}\mathbf{X}$ is an average of the input. Estimating the output $\mathbf{Z}_L$ by simply averaging the inputs results in a blurry output, as illustrated in Figure 3.

**Our method.** Different from standard Transformer, our method enforces sparsity in coefficients by applying $\sigma(\cdot) = \text{prox}(\cdot)$ to obtain $\boldsymbol{\alpha}_L = \sigma(\mathbf{X}_L\phi(\mathbf{X})) = \text{prox}(\mathbf{0}) = \mathbf{0}$, which produces

$$\mathbf{Z}_L = \boldsymbol{\alpha}_L\,\psi(\mathbf{X}) = \mathbf{0}. \tag{13}$$

This indicates that no estimation of the target output is made when there is no observation of the input. However, with the coefficient estimation (9), $\boldsymbol{\alpha}_L \leftarrow \boldsymbol{\alpha}_L + \sum_{i=1}^{L-1} \lambda_i\boldsymbol{\alpha}_i$, we avoid a zero estimation of the target coefficients by linearly combining the coefficients of the context tasks, and produce nonzero output,

$$\mathbf{Z}_L = \boldsymbol{\alpha}_L\,\psi(\mathbf{X}) + \sum_{i=1}^{L-1} \lambda_i\boldsymbol{\alpha}_i\,\psi(\mathbf{X}). \tag{14}$$

Without coefficient estimation, neither standard Transformer nor our method yields informative outputs for $\mathbf{Z}_L$. However, by learning $\lambda_i$ and leveraging the accurate reconstruction of context examples by $\mathbf{Z}_i, \forall i = 1, \cdots, L-1$, $\mathbf{Z}_L = \boldsymbol{\alpha}_L\,\psi(\mathbf{X}) + \sum_{i=1}^{L-1} \lambda_i\boldsymbol{\alpha}_i\,\psi(\mathbf{X})$ is capable to generate meaningful outputs that reuse compositional rules from the context tasks. In practice, we observe the sharp and recurrent attention patterns produced by our method, as shown in Figure 3. We provide details of our analysis in Appendix 9.

## 4 Experiments

### 4.1 Symbolic RAVEN

The S-RAVEN dataset [20], detailed in the original paper, is specifically designed to evaluate compositional reasoning. In S-RAVEN, each task is built from a finite set of rule combinations systematically

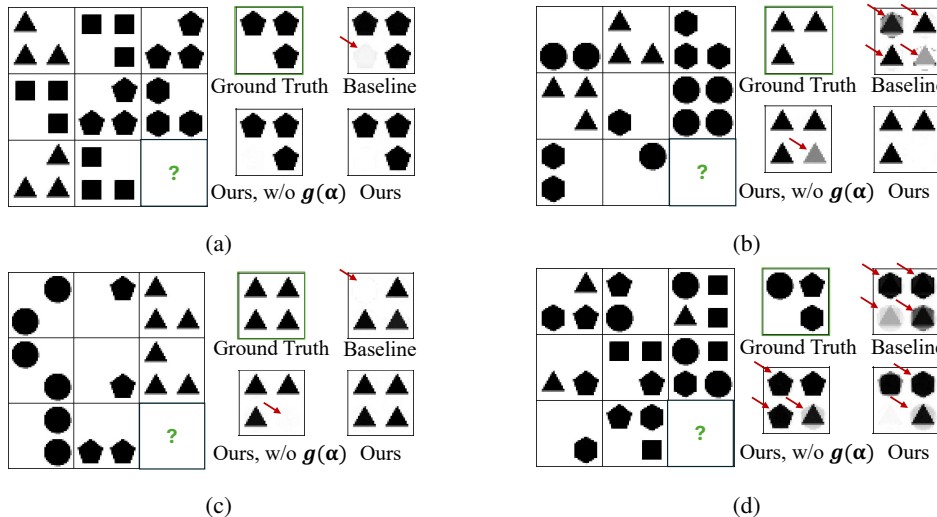

Figure 5: Example results of RAVEN. The model predicts the 9th panel based on the first 8 panels. We compare our method with and without $g(\alpha)$, the coefficient estimation for the target task, alongside the baseline method. The baseline often yields blurry images with incorrect layouts, whereas our method preserves structure and improves compositional accuracy. However, all models occasionally fail on the most challenging cases, *e.g.*, (d).

applied across panels, with each panel represented as a tuple of integers that symbolically encodes its features. Details of the experimental setting are described in Appendix 7.

**Experimental settings.** We train models using the standard decoder-only Transformer architecture and evaluate performance under varying numbers of training tasks and attention layers. Our method builds directly on the S-RAVEN implementation, introducing sparsity into the attention maps and applying a lifting scheme to enhance compositional rule transfer within the attention mechanism.

**Metrics.** To evaluate whether a model trained on a subset of rule combinations can generalize to unseen combinations, we partition all possible rule combinations into separate training and test sets, where 25% of the combinations are held out for testing. Model performance is assessed by measuring the accuracy of correctly predicted examples from the test set.

**Results.** The results, summarized in Table 4, are obtained by running the experiment three times. Our method consistently outperforms baseline approaches, including standard Transformer [26] and HYLA [20], achieving significantly higher accuracy even with fewer layers.

## 4.2 RAVEN dataset

The RAVEN dataset [29] was originally designed for visual reasoning, requiring models to select the correct answer from eight candidates based on the underlying structure of context panels. In contrast, our work focuses on evaluating the compositional capabilities of models by tasking them with generalizing the answer directly, rather than selecting from predefined options—a more challenging objective that demands better understanding and application of the compositional rule.

**Experimental settings.** For our experiments, we adapt the rule framework from RAVEN and focus on the simplified case where examples are arranged in a $2 \times 2$ grid. The model generates the target answer based on the composition of the eight context images. We modify the standard Transformer architecture to serve as a baseline and compare its performance against our approach, which incorporates sparsity and estimation of the coefficients into the attention mechanism.

**Metrics.** To assess model performance, we adapt the Peak Signal-to-Noise Ratio (PSNR) metric to quantify the difference between the generated images and the ground truth. We report the percentage of test samples exceeding PSNR thresholds of 20, 30, and 40, where a higher percentage indicates better reconstruction quality and overall model performance.

**Results.**   As shown in Figure 4, the results demonstrate that our method consistently achieves higher accuracy than the standard Transformer baseline. While the standard Transformer yields nearly 0% of test samples with PSNR above 40, our method maintains around 40%. Additionally, we observe further performance gains when the target coefficient estimation is applied.

Example predictions are visualized in Figure 5, where the baseline model frequently produces blurry images with incorrect arrangements, while our method preserves clear structural information and generates more accurate compositions. Nevertheless, in particularly challenging cases, all models occasionally fail to produce satisfactory outputs.

# 5   Related Work

**In-context compositional learning.**   Recent research on compositional reasoning with transformers has explored several key directions. Some studies focus on understanding and measuring compositional generalization abilities, often identifying gaps between LLM performance on known components and novel compositions, and how these gaps evolve with model scale or in-context learning [7, 9, 16, 21]. Other works delve into the underlying mechanisms and offer explanations for how LLMs achieve or fail at compositional reasoning, for example, by proposing that attention acts as a hypernetwork or by analyzing emergent algorithmic behaviors [15, 19, 20, 24]. Another line of inquiry compares the effectiveness of general pre-training against specialized architectures, investigating whether broad pre-training itself can endow models with strong compositional capabilities, sometimes rivaling or exceeding those of systems explicitly designed for such tasks [2, 8]. However, conventional Transformers often struggle with in-context compositional tasks due to insufficient structural inductive bias. We address this limitation by introducing a sparse coding attention, explicitly designed to capture and transfer structural rules from context examples.

**Sparsity in attention.**   Sparsity has proven to be a powerful principle, and extensive research has investigated its application in Transformers, primarily to reduce the computational complexity [23] of the attention mechanism. Sparse attention mechanisms aim to reduce the number of token pairs being attended to. This includes methods employing fixed, pre-defined sparsity patterns, such as local windowed attention combined with varying forms of global or random attention [1, 3, 28]. Learnable or adaptive sparsity patterns have been explored, where the attention pattern is dynamically determined, for instance, through locality-sensitive hashing [12] or learned routing strategies [18]. Some approaches seek to approximate full attention using kernel methods or low-rank projections, which implicitly reduce computational load without explicit sparse connections [4, 27]. In contrast to prior work, our approach introduces sparsity in attention mainly to enhance the representation of compositional rules. By replacing softmax with soft-thresholding, we promote the learning of structured, localized attention patterns that better capture and encode compositional relationships.

# 6   Conclusion

In this work, we proposed a reformulation of the Transformer architecture to address the challenge of in-context compositional learning. By drawing inspiration from sparse coding, we introduced a framework that represents compositional rules as sparse coefficients over learned dictionaries, enhancing the transferability of structure across tasks. By enforcing sparsity in the coefficients and estimating target coefficients from those of the context tasks, our method further enhances rule transfer and localization within the attention mechanism. Experimental results on in-context compositional learning datasets, such as S-RAVEN and RAVEN benchmark, demonstrate that our approach significantly outperforms standard Transformers, particularly in tasks requiring compositional reasoning and generalization to unseen rule combinations. These findings highlight the potential of combining principles from sparse coding and attention to advance structured reasoning in neural models.

**Limitations.**   While our approach shows promising results in training Transformers on relatively small-scale tasks, its application to large pre-trained models remains unexplored. Although integrating the linear combination of attention maps into pre-trained models could potentially enhance compositional learning, we leave this for future work.

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

# 7 Experimental Details

**The experimental setting of synthetic data.**   We conduct experiments on a synthetic dataset composed of 16 distinct basic elements, which are shown in Figure 6 (a), where each panel is constructed by selecting and combining two of these elements. The examples of training and test data are displayed in Figure 6 (b-c) The model architecture consists of a single-layer Transformer using only self-attention, omitting the feedforward layer. The input sequence length is fixed at $N = 32$, with a feature dimension of 16 and a single attention head ($H = 1$). Training is performed over 200 epochs using a batch size of 128. We optimize the model using the Adam optimizer with a learning rate of 0.001 and employ mean squared error (MSE) loss as the training objective.

**The experimental setting of S-RAVEN.**   We evaluate on the S-RAVEN benchmark [20], where each task is defined by 4 features, sampled from a pool of 8 possible rules. For each task, we generate three input-output sequences of length three, using random inputs for each rule to form the context. Our model architecture follows HYLA [20], varying the number of layers between 4 and 8. The input has a feature dimension of 128 and 16 attention heads ($H = 16$). For baseline comparisons, including HYLA and a standard Transformer, we adopt the original configurations as specified in the HYLA paper. All models use Root Mean Square (RMS) normalization for attention activations. To promote structured representations, our method applies soft thresholding to the attention weights, encouraging sparsity. Training is conducted for one epoch using a batch size of 128, the Adam optimizer with a learning rate of 0.001 and a weight decay of 0.1, and the cross-entropy loss as the objective.

**The experimental setting of RAVEN.**   We conduct experiments on a restricted version of the RAVEN dataset [29], focusing solely on the 2-by-2 grid layout. To ensure deterministic target generation, we remove stochastic variations in rotation and color, so that the target panel is uniquely determined by the eight context panels. Each image is resized to $40 \times 40$ pixels. The model is a standard Transformer with 4 layers, a sequence length of $N = 36$, a feature dimension of 512, and 16 attention heads ($H = 16$). Training is performed over 2000 epochs with a batch size of 256, using the Adam optimizer with a learning rate of 0.0001. The model is trained to minimize mean squared error (MSE) loss.

# 8 Experimental Results

**Sparsity and threshold**   To investigate the impact of the threshold on attention sparsity, we conduct experiments on the RAVEN dataset. Specifically, we measure the average sparsity of the attention maps across all layers, where sparsity is defined as the proportion of zero-valued entries after thresholding. As the threshold increases, more small-magnitude values are suppressed, leading to higher sparsity levels. Our results confirm this trend: larger thresholds consistently yield sparser attention maps, demonstrating the controllable nature of sparsity in our model through the threshold parameter. The results are shown in Table 1.

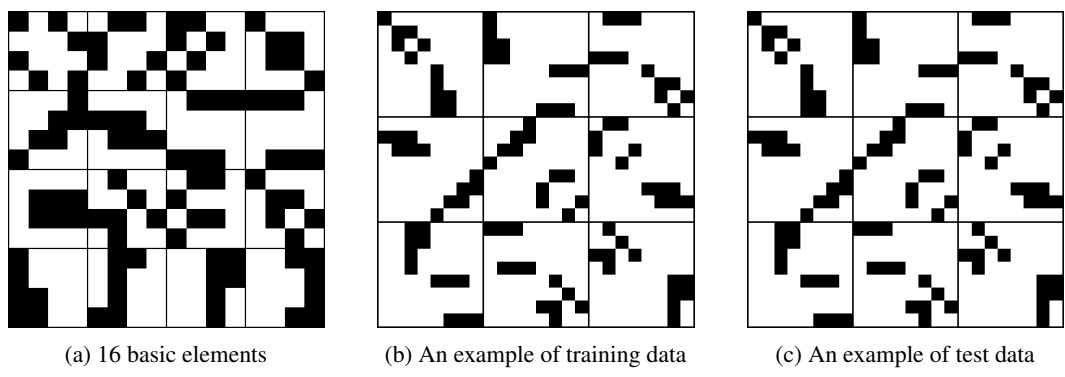

(a) 16 basic elements      (b) An example of training data      (c) An example of test data

Figure 6: Examples of the synthetic dataset.

Table 1: The effect of threshold on the sparsity of the attention map.

| Threshold ($\xi$) | 0.003 | 0.01 | 0.03 | 0.1 | 0.3 |
|---|---|---|---|---|---|
| Sparsity | 18.53 | 57.82 | 90.45 | 97.82 | 99.38 |

Table 2: Variation of basis functions.

| Configs of $\phi(\mathbf{X})$ and $\psi(\mathbf{X})$ | $\mathbf{W}_{qk}^{(h)}\mathbf{X}, \mathbf{W}_{vo}^{(h)}\mathbf{X}$ | $\texttt{ReLU}(\mathbf{W}_{qk}^{(h)}\mathbf{X}), \mathbf{W}_{vo}^{(h)}\mathbf{X}$ | $\mathbf{W}_{qk}^{(h)}\mathbf{X}, \texttt{ReLU}(\mathbf{W}_{vo}^{(h)}\mathbf{X})$ | $\texttt{ReLU}(\mathbf{W}_{qk}^{(h)}\mathbf{X}), \texttt{ReLU}(\mathbf{W}_{vo}^{(h)}\mathbf{X})$ |
|---|---|---|---|---|
| Accuracy | 71.7 | 72.3 | 72.9 | 73.6 |

**Variation of basis functions.**  With the above formulation, we explore different designs for the basis functions $\phi(\cdot)$ and $\psi(\cdot)$ to adjust the expressiveness of models. In the baseline configuration, the basis functions are constructed through linear projections of the input, parameterized by $\mathbf{W}_{qk}^{(h)}$ or $\mathbf{W}_{vo}^{(h)}$. A simple variation is to introduce nonlinearity into the basis construction by applying an activation function, such as $\texttt{ReLU}$, after the linear projections. For instance, a different basis function can be redefined as $\phi(\mathbf{X}) = \texttt{ReLU}(\mathbf{W}_{qk}^{(h)}\mathbf{X})$ or $\psi(\mathbf{X}) = \texttt{ReLU}(\mathbf{W}_{vo}^{(h)}\mathbf{X})$. Incorporating nonlinearity into the basis functions can increase the representational capacity, enabling the model to capture more complex localized patterns beyond those achievable with purely linear projections.

We conduct experiments on the S-RAVEN dataset using a 4-layer Transformer architecture, training the model on a dataset of 20 million samples. We compare the different designs of $\phi(\mathbf{X})$ and $\psi(\mathbf{X})$ by adding the $\texttt{ReLU}$. The results are shown in Table 2.

**Application to language modeling tasks.**  While our primary focus is to address specific limitations of attention mechanisms in compositional tasks, we have conducted experiments on language models to demonstrate our method's effectiveness on standard benchmarks.

We integrate the proposed sparse-coding inspired attention into the Llama-7B [25] model. We then fine-tuned these modified models on several widely-used commonsense reasoning benchmarks and compared the results against both the original base models and those fine-tuned using LoRA/DoRA [10, 13].

In our implementation, we target the model's Attention blocks. We treat the original attention weights (denoted as $\psi(\mathbf{X})$ and $\phi(\mathbf{X})$) as fixed and introduce our core components: new, learnable parameters for sparsity ($\xi$) and the coefficient transfer mechanism ($\lambda_i$). We initialize these new parameters to zero, ensuring that our module has no impact on the model's output before training. By fine-tuning only these new parameters, we can cleanly measure the influence of our method.

Our findings show that models incorporating our model achieve a notable performance improvement over the base Llama model, which is shown in Table 3. Although these results do not yet surpass those from LoRA and DoRA fine-tuning, it's important to consider that our approach uses significantly fewer trainable parameters (over a hundred vs. over 50 million) and has not undergone extensive hyperparameter optimization. The performance gains over the base models suggest that large language models benefit from our mechanism on reasoning tasks, providing compelling evidence of its value. We believe that with further refinement, our approach has the potential to achieve better performance on language modeling tasks. We see the exploration of its application to other benchmarks, such as translation and summarization, as a promising direction for future work.

**Evaluate the models on Im-promptu benchmark.**  We evaluate our method on the Im-promptu benchmark [5], including 3D Shapes, BitMoji Faces, and CLEVr Objects datasets. For this comparison, we adopt the Object-Centric Learner from the original paper as our baseline and integrate our approach by modifying its attention layer. As detailed in the Table 4, our method consistently achieves a lower MSE, demonstrating an improvement over the baseline.

Table 3: Results on language modeling tasks.

| Model | Params | BoolQ | PIQA | HellaSwag | WinoGrande | ARC-c | OBQA | Avg. |
|-------|--------|-------|------|-----------|------------|-------|------|------|
| Llama-7B | - | 56.5 | 79.8 | 76.1 | 70.1 | 63.2 | 77.0 | 70.5 |
| + Ours | 128 | 57.3 | 80.7 | 80.6 | 71.1 | 64.2 | 77.6 | 71.9 |
| LoRA | 55.9M | 67.5 | 80.8 | 83.4 | 80.4 | 62.6 | 79.1 | 75.6 |
| LoRA + Ours | 55.9M | 69.5 | 81.8 | 81.6 | 80.8 | 65.1 | 79.0 | 76.3 |
| DoRA | 56.6M | 69.7 | 83.4 | 87.2 | 81.0 | 66.2 | 79.2 | 77.8 |
| DoRA + Ours | 56.6M | 70.0 | 83.6 | 87.3 | 81.2 | 67.4 | 78.9 | 78.1 |

Table 4: Results on Im-promptu benchmark.

| MSE | 3D Shapes | BitMoji Faces | CLEVr Objects |
|-----|-----------|---------------|---------------|
| OCL | 4.36 | 4.77 | 37.54 |
| Ours | 4.31 | 4.42 | 36.23 |

# 9 Additional Analysis

By representing an input $\mathbf{X}$ as $[\mathbf{X}_1, \cdots, \mathbf{X}_L]^\mathsf{T}$, where $\mathbf{X}_i, \forall i = 1, \cdots, L-1$ and $\mathbf{X}_L \in \mathbb{R}^{\frac{N}{L} \times d}$ are corresponding to context tasks and the target task, we have,

$$\begin{bmatrix} \mathbf{Z}_1 \\ \vdots \\ \mathbf{Z}_L \end{bmatrix} = \begin{bmatrix} \sigma(\mathbf{X}_1 \, \phi(\mathbf{X})) \, \psi(\mathbf{X}) \\ \vdots \\ \sigma(\mathbf{X}_L \, \phi(\mathbf{X})) \, \psi(\mathbf{X}) \end{bmatrix} = \begin{bmatrix} \boldsymbol{\alpha}_1 \, \psi(\mathbf{X}) \\ \vdots \\ \boldsymbol{\alpha}_L \, \psi(\mathbf{X}) \end{bmatrix}. \tag{15}$$

We set $\mathbf{X}_L = \mathbf{0}$, where $\mathbf{0} \in \mathbb{R}^{\frac{N}{L} \times d}$ is a matrix with all zeros, since no observation for the target task.

**Our method.** Different from standard Transformer, our method enforces sparsity in coefficients by applying $\sigma(\cdot) = \text{prox}(\cdot)$ to obtain $\boldsymbol{\alpha}_L = \sigma(\mathbf{X}_L \phi(\mathbf{X})) = \text{prox}(\mathbf{0}) = \mathbf{0}$, which produces

$$\mathbf{Z}_L = \boldsymbol{\alpha}_L \, \psi(\mathbf{X}) = \mathbf{0}. \tag{16}$$

This indicates that no estimation of the target output is made when there is no observation of the input. However, with the coefficient estimation (9), $\boldsymbol{\alpha}_L \leftarrow \boldsymbol{\alpha}_L + \sum_{i=1}^{L-1} \lambda_i \boldsymbol{\alpha}_i$, we avoid a zero estimation of the target coefficients by linearly combining the coefficients of the context tasks, and produce nonzero output,

$$\mathbf{Z}_L = \boldsymbol{\alpha}_L \, \psi(\mathbf{X}) + \sum_{i=1}^{L-1} \lambda_i \boldsymbol{\alpha}_i \, \psi(\mathbf{X}). \tag{17}$$

Without coefficient estimation, neither standard Transformer nor our method yields informative outputs for $\mathbf{Z}_L$. However, by learning $\lambda_i$ and leveraging the accurate reconstruction of context examples by $\mathbf{Z}_i, \forall i = 1, \cdots, L-1$, $\mathbf{Z}_L = \boldsymbol{\alpha}_L \, \psi(\mathbf{X}) + \sum_{i=1}^{L-1} \lambda_i \boldsymbol{\alpha}_i \, \psi(\mathbf{X})$ is capable to generate the target outputs that reuse compositional rules from the context tasks.

## 9.1 Compositional Reconstruction of the Target Output

We have a dictionary of basis elements, $\psi(\mathbf{X}) = \{\psi_j\}_{j=1}^N$. Each output $\mathbf{Z}_i$ for $i = 1, \ldots, L$ is expressed as a linear combination of elements in $\psi(\mathbf{X})$ using coefficient vectors $\boldsymbol{\alpha}_i \in \mathbb{R}^N$, i.e.,

$$\mathbf{Z}_i = \sum_{j=1}^n \boldsymbol{\alpha}_i^{(j)} \psi_j = \boldsymbol{\alpha}_i^\mathsf{T} \psi, \tag{18}$$

where $\psi = [\psi_1, \ldots, \psi_N]^\mathsf{T}$.

**Assumption 9.1.** *The dictionary $\psi(\mathbf{X})$ is **sufficient** to represent the target output $\mathbf{Z}_L$.*

**Assumption 9.2.** *Each of the $L-1$ outputs $\mathbf{Z}_1, \ldots, \mathbf{Z}_{L-1}$ is correctly constructed using coefficient vectors $\boldsymbol{\alpha}_1, \ldots, \boldsymbol{\alpha}_{L-1}$.*

**Assumption 9.3.** *Across* $\{\mathbf{Z}_1, \ldots, \mathbf{Z}_{L-1}\}$, *every dictionary element* $\psi_j$ *is used at least once, i.e.,* $\forall j$, *there exists* $i$ *such that* $\boldsymbol{\alpha}_i^{(j)} \neq 0$.

**Proposition 9.4.** *There exists a set of weights* $\lambda_1, \ldots, \lambda_{L-1}$ *such that:*

$$\boldsymbol{\alpha}_L = \sum_{i=1}^{L-1} \lambda_i \boldsymbol{\alpha}_i, \tag{19}$$

*and* $\boldsymbol{\alpha}_L$ *reconstructs* $\mathbf{Z}_L$ *using only elements in* $\psi(\mathbf{X})$.

*Proof.* Let $\mathcal{A} = \{\boldsymbol{\alpha}_1, \ldots, \boldsymbol{\alpha}_{L-1}\} \subset \mathbb{R}^N$ denote the set of known coefficient vectors. Let $V = \text{span}(\mathcal{A}) \subseteq \mathbb{R}^N$ be the subspace spanned by them. Since from Assumption 9.2, each $\boldsymbol{\alpha}_i$ reconstructs $\mathbf{Z}_i$ correctly and the union of their support covers all dictionary elements, the span $V$ includes directions along all dictionary elements used for constructing $\mathbf{Z}_L$.

From Assumption 9.1, we know there exists some $\boldsymbol{\alpha}_L^* \in \mathbb{R}^N$ such that:

$$\mathbf{Z}_L = \boldsymbol{\alpha}_L^{*\mathsf{T}} \psi. \tag{20}$$

Because $\text{supp}(\boldsymbol{\alpha}_L^*) \subseteq \bigcup_{i=1}^{L-1} \text{supp}(\boldsymbol{\alpha}_i)$, *i.e.*, the dictionary elements needed for $\mathbf{Z}_L$ have already been used in $\mathcal{A}$, and all such directions are already present in $V$, it follows that:

$$\boldsymbol{\alpha}_L^* \in V. \tag{21}$$

Therefore, there exist scalars $\lambda_1, \ldots, \lambda_{L-1}$ such that:

$$\boldsymbol{\alpha}_L^* = \sum_{i=1}^{L-1} \lambda_i \boldsymbol{\alpha}_i.$$

Thus, by setting $\boldsymbol{\alpha}_L := \sum_{i=1}^{L-1} \lambda_i \boldsymbol{\alpha}_i$, we obtain the desired coefficient vector such that:

$$\mathbf{Z}_L = \boldsymbol{\alpha}_L^{\mathsf{T}} \psi.$$

$\square$

Given that the dictionary is sufficient, and the $L-1$ outputs collectively utilize all necessary dictionary elements, the coefficient vector for the $L$-th output can be expressed as a linear combination of previous coefficient vectors. This demonstrates the ability to transfer compositional rules from context examples to new tasks via linear combination of coefficients.

## 10 Computational Resource

We conducted development and experiments on a Linux workstation equipped with a single NVIDIA A5000 GPU (24GB memory). A single run of the synthetic task typically takes 3–5 minutes, while a single S-RAVEN experiment run takes between 60 and 200 minutes. For RAVEN experiments, a full run requires approximately 200 minutes.

