# OpenReview forum: "In-Context Compositional Learning vis Sparse Coding Transformer"
_NeurIPS.cc/2025/Conference — NeurIPS 2025 poster_

### Official Review · Reviewer_i17N · 2025-06-18

**Clarity:** 2
**Significance:** 3
**Originality:** 3
**Rating:** 4
**Confidence:** 4

**Summary:**

The paper proposes a drop-in replacement of the attention mechanism motivated by sparse-coding. The proposed mechanism replaces the softmax with soft-thresholding. The authors rename the attention as coefficients. Given a context of L tokens, the coefficient of token at position L+1 may be defined as a weighted average of the coefficients of the prior token. The authors demonstrate their mechanism in toy tasks that demonstrate compositional generation.

**Questions:**

The key concerns are listed above.

**Ethical Concerns:**

["NO or VERY MINOR ethics concerns only"]

**Final Justification:**

Experiments on fine-tuning addressed my major concern.

**Limitations:**

Yes

**Quality:**

2

**Strengths And Weaknesses:**

The authors study the problem of compositional via sparse-coding. The idea of using sparsity to induce structure and improve generalization is well-motivated in the literature.

However, this article have several issues as it currently stands.

The authors assert strong claim which are not backed by citations or evidences. For example, the authors assert that the softmax function produces dense attention weights [..., which] hinders the model from representing compositional structure inherent in contextual tasks." However, there are several example of transformers that use soft-attention and can generalize outside of its training distributions. I can use any LLMs and make them exhibit systematic generalization.

The paper lacks clarity. For example,
- the authors re-use the symbol $\sigma$ for both the softmax and the "soft-thresholding" function,
- the authors defines $l$ as a task-specific loss function. However, $l$ is not indexed by the task index. Is it really task-specific? This is ambiguous.

The paper is limited to compositional tasks and does not demonstrate the proposed mechanism on standard benchmarks. To be convincing, the authors should compare the method on standard benchmarks (e.g. translation tasks, summarization, etc.) and they should compare against baseline numbers published in the literature. Without such comparison, it is hard to assess the potential impact of the proposed mechanism.

---

> ### Author Rebuttal · Authors · 2025-07-31
>
> Thank you for the constructive comments. We will incorporate this feedback into our revised work.
>
> **Q1. The softmax attention hinders compositional generalization is an unsubstantiated argument.**
>
>
> A: We acknowledge that many powerful models using standard softmax attention can achieve systematic generalization. Our central argument is that explicitly introducing sparsity into the attention mechanism provides distinct advantages for representing compositional structure. While dense attention can learn these relationships implicitly, our work investigates whether adding an explicit inductive bias for sparsity can lead to a more robust or interpretable model of compositional rules. The claim is supported by both a formal analysis and empirical evidence within this specific problem setting:
>
> 1. In Section 3.2, we provide a mathematical breakdown of what happens when a standard Transformer must predict a target based on context examples with no prior observation of the target itself (i.e., a masked input). In this scenario, the softmax operation on a zero input produces a uniform, dense attention map. This forces the model to create the output for the target task by simply averaging the features from the context tasks, leading to the "blurry output" seen in our experiments. Our proposed method, which uses a sparsity-promoting function, explicitly avoids this failure mode.
>
> 2. The experiments in Figure 3 directly support our claim. The baseline model, using standard softmax attention, fails to generalize to a novel compositional rule and produces blurry, averaged outputs. In contrast, our model, under the exact same conditions, successfully generalizes and produces a sharp, correct output. This comparison highlights a clear limitation of dense attention for this type of task.
>
> 3. While LLMs are powerful, the challenge of compositional generalization is not entirely solved. As noted in our related work section, several studies have identified gaps between LLM performance on known components and novel compositions.
>
> We will revise the manuscript to clarify that our work aims to demonstrate the benefits of adding sparsity, rather than arguing that standard softmax attention is inherently flawed.
>
>
>
>
> **Q2. Clarification: (a) The authors reuse the symbol for both the softmax and the "soft-thresholding" function. (b) The definition of a task-specific loss function is ambiguous.**
>
> A: (a) Our intention was to use $\sigma$ as a general placeholder for the non-linear activation function that processes the pre-computed coefficients. In the standard Transformer, this function is softmax. In our proposed model, this function is a sparsity-promoting one, such as $prox(x)$ (soft-thresholding). We will clarify that in our revised paper.
>
> (b) The term "task-specific" was intended to convey that the form of the loss function $l$ is chosen to be appropriate for the overall problem domain being addressed (e.g., image reconstruction for the RAVEN experiments), which is not indexed by the task index. Within a given set of experiments (e.g., all RAVEN tasks), the same loss function—MSE in our case —is used for all individual tasks.
> Therefore, the loss $l(f(C), x_L)$ is dependent on the specific task instance because its arguments (the model's prediction and the ground truth) are determined by that task, even though the mathematical form of the loss function itself remains fixed across all tasks drawn from that distribution. We will revise the wording to "a loss function appropriate for the task" to make this clearer.
>
> **Q3. Application to language modeling tasks.**
>
>
> A: While our paper's primary focus was to address specific limitations of attention mechanisms in compositional tasks, we have conducted new experiments on language models to demonstrate our method's effectiveness on standard benchmarks.
>
> We integrated our proposed sparse-coding inspired attention into the Llama-7B [1] and Llama-2-7B [2] models. We then fine-tuned these modified models on several widely-used commonsense reasoning benchmarks and compared the results against both the original base models and those fine-tuned using LoRA/DoRA [3, 4].
>
> Our findings show that models incorporating our model achieve a notable performance improvement over the base Llama and Llama-2 models. Although these results do not yet surpass those from LoRA and DoRA fine-tuning, it's important to consider that our approach uses significantly fewer trainable parameters (over a hundred vs. over 50 million) and has not undergone extensive hyperparameter optimization. The performance gains over the base models suggest that large language models benefit from our mechanism on reasoning tasks, providing compelling evidence of its value. We believe that with further refinement, our approach has the potential to achieve better performance on language modeling tasks. We see the exploration of its application to other benchmarks, such as translation and summarization, as a promising direction for future work.
>
> | Model | Params | BoolQ | PIQA | HellaSwag | WinoGrande | ARC-c | OBQA | Avg. |
> |:------------:|:------:|:--------:|:------:|:-------:|:----------:|:----------:|:----------:|:----------:|
> | Llama-7B | - | 56.5 | 79.8 | 76.1 | 70.1 | 63.2 | 77.0 | 70.5 |
> | Llama-7B (Ours) | 128 | 57.3 | 80.7 | 80.6 | 71.1 | 64.2 | 77.6 | 71.9 |
> | LLama-7B-LoRA | 55.9M | 67.5 | 80.8 | 83.4 | 80.4 | 62.6 | 79.1 | 75.6 |
> | LLama-7B-DoRA | 56.6M | 69.7 | 83.4 | 87.2 | 81.0 | 66.2 | 79.2 | 77.8 |
> | Llama-2-7B | - | 57.4 | 78.8 | 77.2 | 69.2 | 63.9 | 78.6 | 70.9 |
> | Llama-2-7B (Ours) | 128 | 58.8 | 79.3 | 80.4 | 73.2 | 64.1 | 80.0 | 72.6 |
> | LLama-2-7B-LoRA | 55.9M | 69.8 | 79.9 | 83.6 | 82.6 | 64.7 | 81.0 | 76.9 |
> | LLama-2-7B-DoRA | 56.6M | 71.8 | 83.7 | 89.1 | 82.6 | 68.2 | 82.4 | 79.6 |
>
>
> **Experiment Details:**
> In our implementation, we target the model's Attention blocks. We treat the original attention weights (denoted as $\phi(X)$ and $\psi(X)$) as fixed and introduce our core components: new, learnable parameters for sparsity ($\xi$) (equation (4) in our paper) and the coefficient transfer mechanism ($\lambda_i$) (equation (9) in our paper). We initialize these new parameters to zero, ensuring that our module has no impact on the model's output before training. By fine-tuning only these new parameters, we can cleanly measure the influence of our method.
>
>
> [1] Touvron, Hugo, Thibaut Lavril, Gautier Izacard, Xavier Martinet, Marie-Anne Lachaux, Timothée Lacroix, Baptiste Rozière et al. "Llama: Open and efficient foundation language models." arXiv (2023).
>
> [2] Touvron, Hugo, Louis Martin, Kevin Stone, Peter Albert, Amjad Almahairi, Yasmine Babaei, Nikolay Bashlykov et al. "Llama 2: Open foundation and fine-tuned chat models." arXiv (2023).
>
> [3] Hu, Edward J., Yelong Shen, Phillip Wallis, Zeyuan Allen-Zhu, Yuanzhi Li, Shean Wang, Lu Wang, and Weizhu Chen. "Lora: Low-rank adaptation of large language models." ICLR (2022).
>
> [4] Liu, S.Y., Wang, C.Y., Yin, H., Molchanov, P., Wang, Y.C.F., Cheng, K.T. and Chen, M.H., Dora: Weight-decomposed low-rank adaptation. ICML (2024).

---

> > ### Comment · Reviewer_i17N · 2025-08-03
> > **Response**
> >
> > Thank you for running the experiments on Llama,
> >
> > Your fine-tuning model have much less capacity than LoRA. It would be great if the authors could add their proposed sparse attention with LoRA additional parameters since the added number of parameters of the proposed method is marginal.

---

> > > ### Author Response · Authors · 2025-08-04
> > >
> > > Thank you for the suggestion. Our method can be combined with LoRA and DoRA because they are orthogonal. Our approach modifies coefficients in the Attention block, while LoRA and DoRA perform low-rank updates to the model's weights.
> > >
> > > To validate this, we conducted a preliminary experiment on LLaMA. The results, presented in the table below, indicate that integrating our method leads to a notable improvement in accuracy with a negligible increase in parameters, enhancing the in-context learning capabilities of LoRA or DoRA.
> > >
> > > | Model | Params | BoolQ | PIQA | HellaSwag | WinoGrande | ARC-c | OBQA | Avg. |
> > > |:------------:|:------:|:--------:|:------:|:-------:|:----------:|:----------:|:----------:|:----------:|
> > > | Llama-7B | - | 56.5 | 79.8 | 76.1 | 70.1 | 63.2 | 77.0 | 70.5 |
> > > | Llama-7B (Ours) | 128 | 57.3 | 80.7 | 80.6 | 71.1 | 64.2 | 77.6 | 71.9 |
> > > | LLama-7B-LoRA | 55.9M | 67.5 | 80.8 | 83.4 | 80.4 | 62.6 | 79.1 | 75.6 |
> > > | LLama-7B-LoRA + Ours | 55.9M | 69.5 | 81.8 | 81.6 | 80.8 | 65.1 | 79.0 | 76.3 |
> > > | LLama-7B-DoRA | 56.6M | 69.7 | 83.4 | 87.2 | 81.0 | 66.2 | 79.2 | 77.8 |
> > > | LLama-7B-DoRA + Ours | 56.6M | 70.0 | 83.6 | 87.3 | 81.2 | 67.4 | 78.9 | 78.1 |

---

### Official Review · Reviewer_Qnor · 2025-06-24

**Clarity:** 4
**Significance:** 2
**Originality:** 3
**Rating:** 4
**Confidence:** 1

**Summary:**

The paper adopts a sparse-codes stance on ICL compositional generalisation. The authors reformulate the Transformer attention module so that the latent rule is expressed through sparse weights over a learned dictionary of "atoms". They show the proposed method is more parameter-efficient and slightly outperforms a hyper-network-based baseline. Experiments are conducted solely on the symbolic RAVEN dataset.

**Questions:**

Please see the Weaknesses

**Ethical Concerns:**

["NO or VERY MINOR ethics concerns only"]

**Final Justification:**

After reading other reviews and the rebuttal from the authors, who presented more evidence of the effectiveness of their method across diverse datasets and models, I decided to maintain my score still leaning towards acceptance.

**Limitations:**

yes

**Quality:**

3

**Strengths And Weaknesses:**

### Strengths
- The paper is very well written.
- The paper is accompanied by code.
- The proposed method appears to be more parameter- and data-efficient than HYLA baseline.

### Weaknesses
- **Lack of a theoretical argument.**
  The paper would benefit from a formal treatment of the problem, discussing when this type of generalisation is achievable and how the proposed method advances toward it.

- **Limited experimental evaluation.**
  I acknowledge the authors statement in the checklist that "our paper […] doesn't have a clear impact on real-world applications". Nevertheless, the evaluation would benefit from experiments on more complex data beyond RAVEN. It remains unclear whether the learned dictionary can scale or generalise to a mixture of tasks (e.g., training on both RAVEN and Multi-dSprites) or to an arbitrary number of ICL demonstrations.

- **Missing connection to object-centric work?**
  There is a large body of object-centric learning literature, with (I believe) a direct link between the concept of "atoms" and "slots". Some work explicitly frames compositional generalisation via "local sparsity patterns in Jacobians" with proposed attention regualrization [1, 2]. Because RAVEN dataset can be viewed as an example of higher-order interactions across context examples, the paper would benefit from distinguishing itself from OCL work and discussing existing approaches to task learning in that area.

[1] Brady et al., *Interaction Asymmetry: A General Principle for Learning Composable Abstractions*
[2] Brady et al., *Provably Learning Object-Centric Representations*

---

> ### Author Rebuttal · Authors · 2025-07-31
>
> Thank you for the constructive comments. We will incorporate this feedback into our revised work.
>
> **Q1. Provide theoretical justification for the generalization of the method.**
>
> A: We provide a formal analysis in Section 3.2 that grounds our method's success. We demonstrate mathematically that for unseen targets, standard softmax attention produces a blurry average, whereas our coefficient transfer mechanism, g($\alpha$), can precisely construct the solution by reusing rules from the context. This advantage is further supported by a proof in the appendix, which guarantees a correct solution when the target information is available in the context. We agree that expanding this specific analysis into a more general theory is an important direction for future work.
>
> **Q2. Application to more complex data.**
>
> A: We validated our method's applicability by evaluating its compositional generalization on the Im-promptu benchmark and its performance in language modeling applications.
>
> **Im-promptu benchmark:** Including 3D Shapes, BitMoji Faces, and CLEVr Objects datasets [1]. For this comparison, we adopted the Object-Centric Learner from the original paper as our baseline and integrated our approach by modifying its attention layer. As detailed in the table below, our method consistently achieves a lower MSE, demonstrating an improvement over the baseline.
>
>
> | MSE      | 3D Shapes | BitMoji Faces | CLEVr Objects |
> |----------|-----------|---------------|---------------|
> | OCL | 4.36      | 4.77          | 37.54         |
> | Ours     | **4.31**      | **4.42**          | **36.23**         |
>
> **Language Model**: We tested this approach on language benchmarks, including commonsense reasoning (PIQA, HellaSwag, WinoGrande) and in-context question-answering (BoolQ, ARC-c, OBQA) tasks.
>
> | Model | Params | BoolQ | PIQA | HellaSwag | WinoGrande | ARC-c | OBQA | Avg. |
> |:------------:|:------:|:--------:|:------:|:-------:|:----------:|:----------:|:----------:|:----------:|
> | Llama-7B | - | 56.5 | 79.8 | 76.1 | 70.1 | 63.2 | 77.0 | 70.5 |
> | Llama-7B (Ours) | 128 | 57.3 | 80.7 | 80.6 | 71.1 | 64.2 | 77.6 | 71.9 |
> | LLama-7B-LoRA | 55.9M | 67.5 | 80.8 | 83.4 | 80.4 | 62.6 | 79.1 | 75.6 |
> | LLama-7B-DoRA | 56.6M | 69.7 | 83.4 | 87.2 | 81.0 | 66.2 | 79.2 | 77.8 |
> | Llama-2-7B | - | 57.4 | 78.8 | 77.2 | 69.2 | 63.9 | 78.6 | 70.9 |
> | Llama-2-7B (Ours) | 128 | 58.8 | 79.3 | 80.4 | 73.2 | 64.1 | 80.0 | 72.6 |
> | LLama-2-7B-LoRA | 55.9M | 69.8 | 79.9 | 83.6 | 82.6 | 64.7 | 81.0 | 76.9 |
> | LLama-2-7B-DoRA | 56.6M | 71.8 | 83.7 | 89.1 | 82.6 | 68.2 | 82.4 | 79.6 |
>
> Our method delivers a notable performance improvement over the original Llama [2] and Llama-2 [3] models. While not yet surpassing fune-tuning methods such as LoRA [4] or DoRA [5], our approach is significantly more parameter-efficient and has been achieved without extensive hyperparameter tuning. We believe that with further refinement, our approach has the potential to achieve better performance on language modeling tasks.
>
> [1] Dedhia, Bhishma, Michael Chang, Jake Snell, Tom Griffiths, and Niraj Jha. "Im-promptu: in-context composition from image prompts." NeurIPS (2023).
>
> [2] Touvron, Hugo, Thibaut Lavril, Gautier Izacard, Xavier Martinet, Marie-Anne Lachaux, Timothée Lacroix, Baptiste Rozière et al. "Llama: Open and efficient foundation language models." arXiv (2023).
>
> [3] Touvron, Hugo, Louis Martin, Kevin Stone, Peter Albert, Amjad Almahairi, Yasmine Babaei, Nikolay Bashlykov et al. "Llama 2: Open foundation and fine-tuned chat models." arXiv (2023).
>
> [4] Hu, Edward J., Yelong Shen, Phillip Wallis, Zeyuan Allen-Zhu, Yuanzhi Li, Shean Wang, Lu Wang, and Weizhu Chen. "Lora: Low-rank adaptation of large language models." ICLR (2022).
>
> [5] Liu, S.Y., Wang, C.Y., Yin, H., Molchanov, P., Wang, Y.C.F., Cheng, K.T. and Chen, M.H., Dora: Weight-decomposed low-rank adaptation. ICML (2024).
>
>
>
> **Q3. Discussion of object-centric work.**
>
>
> A:  Both the "dictionary atoms" in our framework and the "slots" in OCL literature serve as learned primitives for composing complex inputs.
>
> Our approach is distinguished from most OCL work by its motivation and mechanism: Our method is explicitly derived from the principles of sparse coding. The goal was to reformulate the entire attention block within a Transformer as a sparse decomposition and reconstruction process. OCL models, while also focused on compositionality, often employ different architectural paradigms built from the ground up, such as iterative refinement or specialized slot-based attention mechanisms.
>
> Our core idea is the replacement of softmax with a sparsity-promoting function ($prox$) and the introduction of an explicit, learnable coefficient transfer scheme (g($\alpha$)). This makes our method a direct modification of the standard Transformer block, aimed at improving its inherent compositional reasoning.
>
> We will incorporate a discussion of OCL in the related work section of a revised version.

---

> > ### Comment · Reviewer_Qnor · 2025-08-05
> > **A Reply to the Rebuttal**
> >
> > After reading other reviews and the rebuttal from the authors, who presented more evidence of the effectiveness of their method across diverse datasets and models, I decided to maintain my score still leaning towards acceptance.

---

> > > ### Author Response · Authors · 2025-08-07
> > >
> > > We sincerely thank the reviewer for your time and constructive feedback.

---

### Official Review · Reviewer_Dgmp · 2025-06-30

**Clarity:** 4
**Significance:** 3
**Originality:** 2
**Rating:** 5
**Confidence:** 3

**Summary:**

The paper builds on sparse coding to propose a new attention mechanism by which transformers perform better at compositional tasks. Precisely the authors propose to reformulate the attention mechanism as an encoding and a decoding dictionary. The encoding dictionary transforms an input into a sparse set of compositional endoding atoms, while the decoding dictionary uses the previous coefficients to recombine decoding atoms, generating an output. At test time, the coefficients are updated by taking into account  both the coefficients of the target task and of context tasks.

The paper asesses the method by evaluating it first on a toy example and on the RAVEN dataset, both of them being examples of in-context compositional tasks. Overall, their proposal outperforms the standard Transformer architecture.

**Questions:**

Why is prox(x) used instead of training with L1  regularization on the coefficient values?
How do the results change as a function of atoms in both dictionaries? And as a function of dimensionality?

**Ethical Concerns:**

["NO or VERY MINOR ethics concerns only"]

**Final Justification:**

The paper, while similar to HYLA, proposes an interesting interpretation of self-attention including sparsity as an inductive bias.

**Limitations:**

yes

**Quality:**

3

**Strengths And Weaknesses:**

## Strengths

- The paper is well-written, the motivation, goal and results are clearly conveyed
- The method is simple and provides with an interesting reformulation of self-attention
- The proposal shows a remarkable improvement with respect to the baseline Transformer architecture

## Weaknesses

- Not clear why the toy example is relevant, it seems to me like it is redundant with RAVEN.
- The work is very similar to "Attention as a Hypernetwork" paper. The paper is cited, but I would like to see a more direct explanation on how both proposals differ.
- No application to language, where a compositional inductive bias could also be beneficial, while showing whether the method scales to bigger network sizes.
- I would like to see more results on atom number and hidden dimensionality, and how these affect performance.

---

> ### Author Rebuttal · Authors · 2025-07-31
>
> Thank you for the insightful feedback. We will incorporate this feedback into our revised work.
>
> **Q1. Clarification of the toy example.**
>
> A: The toy example in Figure 3 serves a distinct and crucial purpose that complements the main RAVEN experiments. While the RAVEN datasets demonstrate the model's performance on complex, established benchmarks, the toy example is a controlled, synthetic environment designed for better illustration and detailed analysis:
>
> 1. It provides an interpretable visualization of the core problem. More importantly, it allows us to test for compositional generalization in a direct way. The model is trained on one compositional rule (Figure 3a) and tested on an unseen rule (Figure 3b). This demonstrates the model's ability to infer and transfer abstract rules, which is the central claim of our paper.
>
> 2. The simplicity of the toy example enables a step-by-step mathematical analysis of why the standard Transformer fails and our method succeeds. As detailed in Section 3.2, we show how the baseline's softmax operation on a masked input leads to an indiscriminate averaging of features, resulting in a *blurry output*. In contrast, our method's use of a sparsity-promoting function ($prox(x)$) and the coefficient transfer mechanism (g($\alpha$)) avoids this issue and produces a sharp, correct output. This level of granular analysis would be difficult to illustrate with the complexity of the full RAVEN dataset.
>
>
> **Q2. Compare with "Attention as a Hypernetwork (HYLA)".**
>
> A: While both approaches reinterpret the attention mechanism, our proposal differs in two fundamental ways that stem from our core inspiration: sparse coding.
>
> 1. The central contribution of our work is the introduction of explicit sparsity into the attention coefficients. We replace the standard softmax function with a sparsity-promoting nonlinearity like soft-thresholding ($prox$). This is a key architectural change designed to produce structured, disentangled representations that capture compositional rules. HYLA does not enforce this type of sparsity. The empirical results in our paper (Table in Figure 4) show our method consistently outperforms HYLA, suggesting that this explicit sparsity is a beneficial inductive bias for these tasks.
>
> 2. We introduce an explicit mechanism, g($\alpha$), to transfer compositional rules from context to target. This step estimates the target coefficients as a learnable linear combination of the context coefficients. This mechanism is directly inspired by the lifting scheme and provides a clear procedure for rule transfer. This explicit transfer step is a unique component of our framework.
>
> **Q3. Application to language model.**
>
> A: We tested this approach on language benchmarks, including commonsense reasoning (PIQA, HellaSwag, WinoGrande) and in-context question-answering (BoolQ, ARC-c, OBQA) tasks. The details of our experimental design are provided below.
>
> | Model | Params | BoolQ | PIQA | HellaSwag | WinoGrande | ARC-c | OBQA | Avg. |
> |:------------:|:------:|:--------:|:------:|:-------:|:----------:|:----------:|:----------:|:----------:|
> | Llama-7B | - | 56.5 | 79.8 | 76.1 | 70.1 | 63.2 | 77.0 | 70.5 |
> | Llama-7B (Ours) | 128 | 57.3 | 80.7 | 80.6 | 71.1 | 64.2 | 77.6 | 71.9 |
> | LLama-7B-LoRA | 55.9M | 67.5 | 80.8 | 83.4 | 80.4 | 62.6 | 79.1 | 75.6 |
> | LLama-7B-DoRA | 56.6M | 69.7 | 83.4 | 87.2 | 81.0 | 66.2 | 79.2 | 77.8 |
> | Llama-2-7B | - | 57.4 | 78.8 | 77.2 | 69.2 | 63.9 | 78.6 | 70.9 |
> | Llama-2-7B (Ours) | 128 | 58.8 | 79.3 | 80.4 | 73.2 | 64.1 | 80.0 | 72.6 |
> | LLama-2-7B-LoRA | 55.9M | 69.8 | 79.9 | 83.6 | 82.6 | 64.7 | 81.0 | 76.9 |
> | LLama-2-7B-DoRA | 56.6M | 71.8 | 83.7 | 89.1 | 82.6 | 68.2 | 82.4 | 79.6 |
>
> Our method delivers a notable performance improvement over the original Llama [1] and Llama-2 [2] models. While not yet surpassing fune-tuning methods such as LoRA [3] or DoRA [4], our approach is significantly more parameter-efficient and has been achieved without extensive hyperparameter tuning. We believe that with further refinement, our approach has the potential to achieve better performance on language modeling tasks.
>
> **Experiment Details:** The scope of this work was to propose and validate our sparse coding-inspired attention mechanism in a domain where the need for compositional reasoning is clear and standard Transformers struggle. The visual reasoning tasks in S-RAVEN and RAVEN provided a controlled environment to establish the effectiveness of our approach.
>
> To evaluate our method on large pre-trained models, we adapt it into a lightweight component for fine-tuning, using LLaMA as our foundation model. Our approach integrates into the existing architecture by adding a new module while keeping the vast majority of the pre-trained weights frozen.
>
> Specifically, we target the model's Attention blocks. We treat the original attention weights (denoted as $\phi(X)$ and $\psi(X)$) as fixed and introduce our core components: new, learnable parameters for sparsity ($\xi$) (equation (4) in our paper) and coefficient recombination ($\lambda_i$) (equation (9) in our paper). We initialize these new parameters to zero, ensuring that our module has no impact on the model's output before training. By fine-tuning only these new parameters, we can measure the influence of our method.
>
> [1] Touvron, Hugo, Thibaut Lavril, Gautier Izacard, Xavier Martinet, Marie-Anne Lachaux, Timothée Lacroix, Baptiste Rozière et al. "Llama: Open and efficient foundation language models." arXiv (2023).
>
> [2] Touvron, Hugo, Louis Martin, Kevin Stone, Peter Albert, Amjad Almahairi, Yasmine Babaei, Nikolay Bashlykov et al. "Llama 2: Open foundation and fine-tuned chat models." arXiv (2023).
>
> [3] Hu, Edward J., Yelong Shen, Phillip Wallis, Zeyuan Allen-Zhu, Yuanzhi Li, Shean Wang, Lu Wang, and Weizhu Chen. "Lora: Low-rank adaptation of large language models." ICLR (2022).
>
> [4] Liu, S.Y., Wang, C.Y., Yin, H., Molchanov, P., Wang, Y.C.F., Cheng, K.T. and Chen, M.H., Dora: Weight-decomposed low-rank adaptation. ICML (2024).
>
>
> **Q4. Effect of atom number and hidden dimensionality on performance.**
>
> A: We conducted a study on the S-RAVEN dataset to analyze the impact of atom count and hidden dimensionality. Our results indicate a direct correlation between these parameters and model accuracy; performance consistently improves as the number of atoms increases from 16 to 128, and likewise, as the hidden dimension expands from 16 to 128. As detailed in the table, this improved accuracy reflects an increase in model capability, which comes at the cost of more parameters per layer.
>
> | Num of atoms | 16    | 32    | 64   | 128   |
> |--------------|-------|-------|------|-------|
> | Accuracy     | 50.12 | 60.67 | 63.1 | 65.68 |
> |Params|35.8K|38.9K|45.1K|57.3K|
>
> | Hidden Dim | 16   | 32   | 64  | 128  |
> |------------|-------|-------|------|-------|
> | Accuracy   | 45.03 | 55.32 | 63.1 | 65.12 |
> |Params|11.3K|22.6K|45.1K|90.2K|
>
> **Q5. Why is prox(x) used instead of training with L1 regularization on the coefficient values?**
>
> A: The proximal operator $prox(x)$ is the mathematical tool we used to solve an optimization problem that includes L1 regularization. We add the L1 penalty ($\lambda ||x||_1$) to our objective function to encourage sparsity, but since this term is non-differentiable everywhere, we can't use standard gradient descent. Instead, we use the proximal operator, where the $prox(x)$ step is the specific operation that correctly applies the L1 penalty by shrinking coefficients and forcing small ones to zero.

---

> > ### Comment · Reviewer_Dgmp · 2025-08-04
> >
> > The authors have addressed most of my concerns, I have updated my score accordingly

---

> > > ### Author Response · Authors · 2025-08-04
> > >
> > > We sincerely thank the reviewer for your time and constructive engagement.

---

### Official Review · Reviewer_1bo9 · 2025-07-01

**Clarity:** 3
**Significance:** 3
**Originality:** 2
**Rating:** 4
**Confidence:** 4

**Summary:**

This paper proposes an improved Transformer architecture for in-context compositional learning, which enhances the compositional generalization ability of Transformers. Specifically, a novel attention mechanism inspired by sparse coding is proposed, where inputs are projected onto an encoding dictionary to produce sparse coefficients that represent compositional rules. These coefficients combine elements from a decoding dictionary to generate the final output. To enable rule transfer, the sparse coefficients of the target task are estimated via a linear combination of the context coefficients. The proposed architecture outperforms standard Transformers on S-RAVEN and RAVEN, demonstrating stronger compositional generalization ability.

**Questions:**

In Figure 3, the baseline model generates relatively blurry images. Would the results change if the MSE loss is replaced with the cross-entropy loss, which is commonly used when training Transformers?

**Ethical Concerns:**

["NO or VERY MINOR ethics concerns only"]

**Final Justification:**

The response has addressed my concerns. The experiments on more datasets or benchmarks improve the applicability of the proposed method, and the authors also indicated that these experiments will be included in the final version. So I raise my score to borderline accept.

**Limitations:**

yes

**Paper Formatting Concerns:**

I think there is no formatting issue that needs to be discussed here.

**Quality:**

2

**Strengths And Weaknesses:**

Strengths

1. This paper identifies a key weakness that Transformers lack the inductive bias to solve compositional generalization tasks. The motivation is practical and attempts to solve an important challenge in current AI research.

2. The attention mechanism based on sparse coding, along with the learnable encoding and decoding dictionaries, provides an intuitive and partially interpretable way to capture discrete compositional rules.

3. The overall architecture is relatively simple, suggesting that the architecture is potentially scalable and extensible.

Weaknesses

My main concerns lie in the evaluation.

1. A key of the architecture is its ability to handle novel rule compositions at test time. However, the RAVEN dataset does not seem to support this setting. As far as I know, in the standard RAVEN configuration, the training and test splits differ only in the values of image attributes, not in the combinations of rules and attributes. This means the rule-attribute compositions in the test set are already seen during training.

2. The PGM dataset [1] may be more appropriate. It includes held-out splits where rule combinations are disjoint between training and testing. I suggest the authors run experiments under the held-out splits of PGM to better validate the methods.

3. The paper lacks discussion and comparison with models that address matrix reasoning tasks. Many existing models, such as neuro-symbolic methods like PrAE [2] and NVSA [3], are capable of in-context compositional learning and compositional generalization in matrix reasoning tasks like RAVEN.

4. [4] discusses compositional generalization in a broader context and introduces a benchmark to evaluate the compositional generalization ability. I recommend the authors compare with the method proposed in [4] and evaluate the models on their benchmark.

[1] Measuring abstract reasoning in neural networks. ICML 2018.

[2] Abstract spatial-temporal reasoning via probabilistic abduction and execution. CVPR 2021.

[3] A neuro-vector-symbolic architecture for solving raven’s progressive matrices. Nature Machine Intelligence, 2023.

[4] Im-promptu: in-context composition from image prompts. NeurIPS 2023.

---

> ### Author Rebuttal · Authors · 2025-07-31
>
> Thank you for the constructive comments. Below are our responses to each of your comments. We will incorporate this feedback into our revised work.
>
> **Q1. The standard RAVEN dataset is inadequate for testing novel rule compositions.**
>
> A: Different from the standard RAVEN dataset, we used two distinct datasets in our paper to evaluate different aspects of compositional learning:
>
> 1. S-RAVEN: As stated in the paper, this dataset is explicitly designed to evaluate compositional reasoning. Our experimental setup for S-RAVEN involved holding out 25% of all possible rule combinations for the test set, ensuring that the model was evaluated on novel compositions not seen during training.
>
> 2. RAVEN: For the visual RAVEN dataset, our primary goal was to evaluate the model's ability to generate the correct answer *directly* from the context images, rather than selecting from a list of candidates. This generative task is inherently more challenging than the standard discriminative format and serves as a strong test of the model's ability to understand and apply the underlying compositional rule, even if the rule structure was seen during training.
>
> We believe this two-pronged approach allowed us to test our model's ability to generalize to both unseen rule structures and to perform a challenging generative task.
>
> **Q2 & Q3. Recommend using the held-out splits from the PGM dataset for more robust validation. Provide discussion of relevant neuro-symbolic models for matrix reasoning, such as PrAE and NVSA.**
>
>
> A: To evaluate our approach, we integrate our coefficient transfer mechanism, g($\alpha$), into the MRNet model [1]. We then compare the performance of our enhanced model against the original MRNet as well as other baselines, including PrAE and NVSA, on both the RAVEN and PGM datasets. The logic and details of our experimental design are provided below.
>
>
> | PGM | Accuracy |
> |--------|----------|
> | MRNet   | 68.34    |
> | NVSA   | 68.30    |
> | Ours   |    **68.92**      |
>
> Our approach demonstrates better performance across both benchmarks. Specifically, it outperforms MRNet and NVSA on the PGM dataset and shows consistent improvement over PrAE and MRNet on the RAVEN dataset.
>
> |   RAVEN   |  Avg  | Center  | 2x2 grid  | 3x3 grid  |  L-R  |  U-D  | O-IC  | O-IG |
> |:----------:|:-----:|:-------:|:---------:|:---------:|:-----:|:-----:|:-----:|:----:|
> | ResNet+DRT | 59.6  |  58.1   |   46.5    |   50.4    | 65.8  | 67.1  | 69.1  | 60.1 |
> |    PrAE    |  60.3 |   70.6  |    83.5   |    30.5   |  88.5 |  89.2 |  37.3 | 22.7 |
> |    NVSA    |  87.7 |   99.7  |    93.5   |    57.1   |  99.8 |  99.7 |  98.6 | 65.4 |
> |    MRNet   |  74.7 |   96.2  |    49.1   |    45.9   |  93.7 |  94.2 |  92.5 | 51.3 |
> |    Ours    |  77.8 |   96.7  |   53.1    |    51.6   |  95.5 |  95.8 |  94.3 | 57.7 |
>
>
> **Experiment Details:** Our model's primary goal is to test the in-context compositional learning capability of a *standard Transformer*. We intentionally kept the architecture simple to isolate this ability, rather than introducing complex, task-specific designs to maximize performance on a specific benchmark.
>
> This focus differs from previous state-of-the-art works on the RAVEN and PGM datasets, which are typically designed for classification. These models often rely on complex reasoning blocks built upon CNN backbones like ResNet. A direct comparison is challenging:
> Simply swapping the CNN backbone in these models with a ViT causes a significant performance drop (e.g., from 57.1% to 43.5% on RAVEN). This is because ViT modules are data-hungry and struggle to generalize on these structured reasoning tasks without extensive pre-training or data.
> Our method is not designed to be a plug-in replacement for the specialized reasoning blocks in those models. While our approach does improve the performance of a vanilla ViT baseline (e.g., from 43.5% to 47.6%), it doesn't close the gap with highly engineered, CNN-based architectures.
>
> To demonstrate the versatility of our coefficient transfer mechanism beyond generative tasks, we adapt it for classification by integrating it into the MRNet baseline. We select MRNet because its multi-head attention model is highly compatible with our approach. The integration is straightforward: we apply our coefficient transfer mechanism, g($\alpha$), directly to the panel-level features extracted by MRNet's panel-processing blocks.
>
>
> [1] Benny, Y., Pekar, N. &Wolf, L. Scale-localized abstract reasoning. CVPR (2021).
>
>
>
>
> **Q4. Evaluate the models on Im-promptu benchmark.**
>
> A: We evaluate our method on the Im-promptu benchmark, including 3D Shapes, BitMoji Faces, and CLEVr Objects datasets. For this comparison, we adopt the Object-Centric Learner from the original paper as our baseline and integrate our approach by modifying its attention layer. As detailed in the table below, our method consistently achieves a lower MSE, demonstrating an improvement over the baseline.
>
>
> | MSE      | 3D Shapes | BitMoji Faces | CLEVr Objects |
> |----------|-----------|---------------|---------------|
> | OCL | 4.36      | 4.77          | 37.54         |
> | Ours     | **4.31**      | **4.42**          | **36.23**         |
>
>
>
> **Q5. Will cross-entropy loss help blurry images of the baseline model?**
>
> A: Yes, using cross-entropy loss results in visually sharper outputs. However, it does not improve the performance on our primary evaluation metric, PSNR. Since PSNR is our quantitative standard for image reconstruction quality, MSE is the more appropriate choice. Cross-entropy loss is most straightforwardly applied to binary or one-hot categorical data. The RAVEN dataset, however, contains non-binary images with grayscale values. Adapting cross-entropy for this type of continuous data is non-trivial and can introduce training instability. Given these considerations, we choose MSE loss as it provides a more stable and suitable objective for the specific data and evaluation metrics used in our paper.

---

> > ### Comment · Reviewer_1bo9 · 2025-08-05
> >
> > Thanks for the detailed response. These results have addressed my earlier concerns. The experiments on more datasets or benchmarks improve the applicability of the proposed method, and the authors also indicated that these experiments will be included in the final version. So I will raise my score accordingly.

---

> > > ### Author Response · Authors · 2025-08-07
> > >
> > > We sincerely thank the reviewer for your time and constructive feedback.

---

### Decision · Program_Chairs · 2025-09-17

**Decision:**

Accept (poster)

**Comment:**

This paper proposes a sparse coding-inspired reformulation of Transformer attention to improve in-context compositional learning, introducing explicit sparsity and a coefficient transfer mechanism. Reviewers appreciated the clear motivation, simplicity of the approach, and promising improvements across symbolic reasoning tasks, vision benchmarks, and even language models, highlighting its potential as a lightweight and interpretable inductive bias. Concerns were raised regarding limited theoretical grounding, initial evaluation scope, and connections to related work, but the rebuttal provided clarifications, additional experiments on diverse datasets and language tasks, and comparisons with relevant baselines, which strengthened confidence in the contribution. While further work on theoretical foundations, scaling, and broader applications would enhance the impact, the consensus is that the paper makes a meaningful and timely contribution. I therefore recommend acceptance.